# Radiative and microphysical responses of clouds to an anomalous increase in fire particles over the Maritime Continent in 2015

Azusa Takeishi [1] and Chien Wang [1]

[1]Laboratoire d'Aérologie, UPS/CNRS, 14 avenue Edouard Belin, 31400 Toulouse, FRANCE

**Correspondence:** A. Takeishi (azusa.takeishi@aero.obs-mip.fr)

**Abstract.** The year of 2015 was an extremely dry year for Southeast Asia where the direct impact of strong El Niño was in play. As a result of this dryness and the relative lack of rainfall, an extraordinary amount of aerosol particles from biomass burning remained in the atmosphere over the Maritime Continent during the fire season. This study uses the Weather Research and Forecasting model coupled with Chemistry to understand the impacts of these fire particles on cloud microphysics and radiation during the peak biomass burning season in September. Our simulations, one with fire particles and the other without them, cover the entire Maritime Continent region at a cloud-resolving resolution (4 km) for the entire month of September in 2015. The comparison of the simulations shows a clear sign of precipitation enhancement by fire particles through microphysical effects; smaller cloud droplets remain longer in the atmosphere to later form ice crystals, and/or they are more easily collected by ice-phase hydrometeors, in comparison to droplets under no fire influences. As a result, mass of ice-phase hydrometeors increases in the simulation with fire particles, so does rainfall. On the other hand, the aerosol radiative effect weakly counteracts the invigoration of convection. Clouds are more reflective in the simulation with fire particles as ice mass increases. Combined with the direct scattering of sunlight by aerosols, the simulation with fire particles shows higher albedo over the simulation domain on average. The simulated response of clouds to fire particles in our simulations clearly differs from what was presented by two previous studies that modeled aerosol-cloud interaction in years with different phases of El Niño–Southern Oscillation (ENSO), suggesting a further need for an investigation on the possible modulation of fire-aerosol-convection interaction by ENSO.

## 1 Introduction

The area of Southeast Asia (SEA) is characterized by the tropical monsoon climate where the rain belt meridionally shifts across the region with season. Over this region, multiple dynamical factors, in addition to the monsoon, are concurrently in play; land-sea breeze on a daily scale due to the contrast in surface heating, the Madden-Julian Oscillation (MJO) on an intraseasonal scale, El Niño–Southern Oscillation (ENSO) on a global scale, and the topographical influence on the flow patterns in general. Xavier et al. (2014), for instance, presented the evidence for the direct impact of the MJO on the probability of extreme rainfall events over SEA, using observational datasets. Meanwhile, a recent summary on the ENSO teleconnection by Lenssen et al. (2020) showed the strong impact of ENSO on the amount of precipitation over SEA by presenting the correlation between El Niño with anomalous drying and La Niña with anomalous wetting. Indeed, the TRMM rainfall data

(available at https://gpm.nasa.gov/data/directory) in Figure 1 seems to confirm this relationship between ENSO and the amount of precipitation over SEA. This relationship can be explained by the zonal shifting of the Walker circulation, which defines ENSO itself; during El Niño, the convective branch of the Walker circulation over the warm pool moves eastwards away from SEA, whereas it gets strengthened near SEA during La Niña (e.g., Wang et al., 2017). Thus, SEA is subject to flow fields and circulation patterns driven by varying scales of atmospheric phenomena.

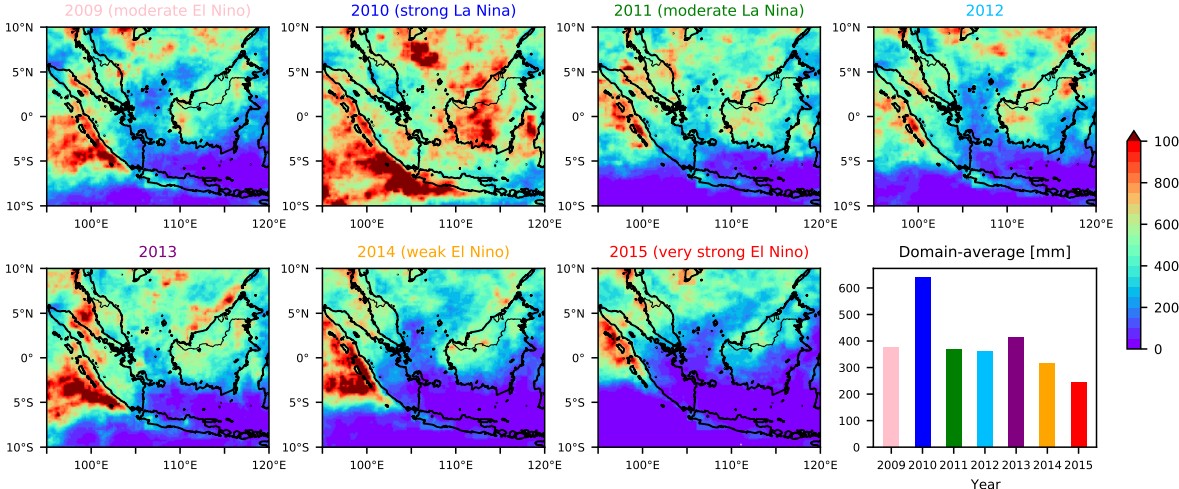

**Figure 1.** 2-month accumulated precipitation [mm] observed by TRMM in September and October of each year from 2009 to 2015, calculated from the monthly mean precipitation rate data (TRMM 3B43). Domain-mean amounts are shown in the bar graph at lower right. The ENSO phases are based on the 3-month running mean Oceanic Niño Index (https://origin.cpc.ncep.noaa.gov/products/analysis_monitoring/ensostuff/ONI_v5.php).

SEA is also characterized by the emissions of aerosol particles from biomass burning activities (hereinafter "fire particles") with a clear seasonal cycle. According to Lin et al. (2014), SEA can be split into the Indochina and the Maritime Continent (MC) based on the peak biomass burning season, which is March and September, respectively. This is confirmed by our analysis on MODIS Aerosol Optical Depth (AOD) data (available at https://ladsweb.modaps.eosdis.nasa.gov/), shown in Figure 2, where the difference in AOD clearly stems from the seasonal meridional shift of the rain belt. In addition to the seasonality, the amount of aerosols over the region is also subject to the interannual variability according to ENSO. Likely because of the tight connection between aerosols and their wet scavenging by rainfall, AOD and the amount of rainfall often show an inverse relationship (e.g., Zhu et al., 2021). Indeed, our analysis of MODIS fire data in Figure 3 also shows an increased (decreased) number of fires during El Niño (La Niña), which confirms the sensitivity of the aerosol abundance in the atmosphere to ENSO.

Microphysical and radiative impacts of fire particles over SEA have been suggested by some observational studies. For instance, Rosenfeld (1999) observed a significant reduction of cloud droplet sizes over the island of Borneo, based on TRMM data, when clouds were downwind of biomass burning. The investigations of two recent field campaigns over the northern part of SEA by Lin et al. (2013) revealed the detailed chemical and radiative characteristics of fire particles, while their impacts

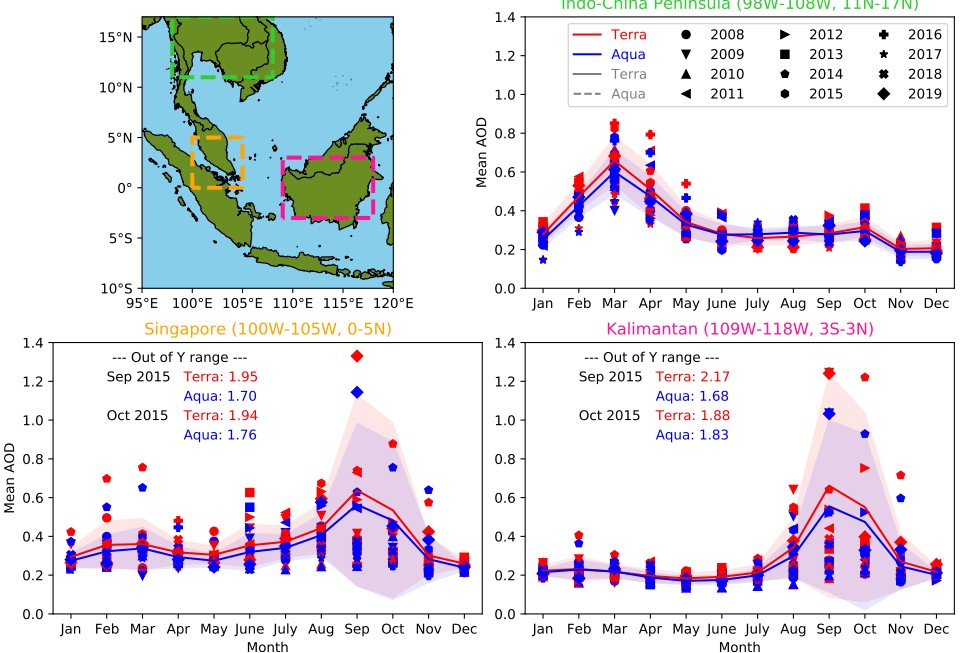

**Figure 2.** MODIS-observed monthly mean AOD averaged over 2008-2019, separately for the three areas shown on the upper left map. The data MOD/MYD08_M3 is 1°× 1°. The shading shows ± standard deviation. The rain belt shifts between ∼20°N (boreal summer) and ∼5°S (boreal winter) around this area (e.g., Schneider et al., 2014).

on clouds remained to be clarified. Similarly, the review study by Tsay et al. (2013) summarized the observational findings on
the characteristics and the seasonality of fire particles over the Indochina. Another review study on aerosols and clouds over
SEA by Lin et al. (2014) pointed out that the largest difficulty lies in the simultaneous observation of clouds and aerosols from
satellites, as aerosol data gets "contaminated" when clouds exist. Therefore, in order to fully understand the connection of fire
particles with cloud characteristics, modeling is indispensable.

     In spite of the abundance of fire particles, only a few modeling studies have focused on aerosol-cloud interactions (ACIs)
over SEA. This may be due to the complexity of multi-scale dynamics over the region that differs quite significantly from
season to season and/or year to year, or potentially because of the practical reason as a cloud-resolving simulation that covers
the entire SEA is computationally expensive. Lee et al. (2014), for example, used the simulations from the GEOS-5 AGCM
model (resolution 2.5°× 2°) to find the reduction of precipitation over SEA due to both indirect and semi-direct effects of
aerosols. Ge et al. (2014), who utilized the Weather Research and Forecasting model (WRF; Skamarock et al., 2008) coupled
with Chemistry (WRF-CHEM; Grell et al., 2005) for finer-resolution (27 km) simulations, found a decrease (increase) in
cloud fraction during daytime (nighttime) due to the cloud radiative effects of aerosols (including the semi-direct effect) that
altered vertical and horizontal flow fields. Simulations in these studies, however, were not on a cloud-resolving scale, which
often refers to a horizontal resolution of ∼4 km or finer that does not require parameterization of convections. Hodzic and

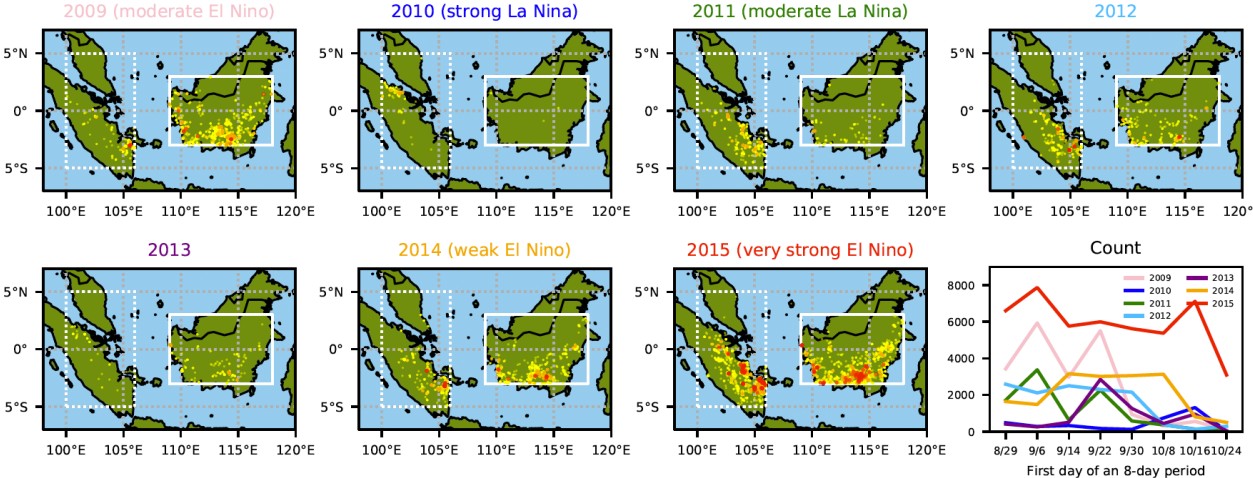

**Figure 3.** Number of fires (with high confidence) within the rectangles, according to 1-km MODIS 8-day fire data (MOD/MYD14A2); yellow represents 2-3 counts, orange 4-5 counts, and red > 5 counts (the maximum is 16, as Terra and Aqua are counted separately for 8 time frames within September-October). The lower right plot shows the total pixel counts of fires with high confidence within the rectangles.

Duvel (2018, hereinafter HD18) ran cloud-resolving simulations using WRF-CHEM over the Borneo island for a 40-day period, including the entire month of September in 2009. According to their simulations, the inclusion of fire particles in the simulations resulted in a reduction of precipitation on island-average, but when particle absorptivity was raised, a nighttime enhancement of precipitation was additionally seen. Thus, they concluded that the response of rainfall to aerosol perturbations depended heavily on the absorptivity of fire particles. In the recent study by Lee and Wang (2020, hereinafter LW20), they also used cloud-resolving WRF-CHEM simulations (resolution 5 km) to investigate the impacts of fires on clouds in SEA over a 4-month period from June to September in 2008. Their simulations included not only the Borneo island but also the Sumatra island and the Malay and Indochina peninsulas. Even though it is one of a few cloud-resolving modeling studies on ACIs over SEA for such a long period of time, changes of seasons within the time period allowed the dominant flow patterns and emissions to change and hence made it difficult to find ACI signals that were consistent for 4 months. From their in-depth analysis on selected cases, they found a reduction of nocturnal rainfall over the western Borneo due mainly to the semi-direct effect of fire particles.

This study aims at further deepening our understanding of ACI over MC during the peak fire season in an extremely dry year of 2015 (Figures 1-3) due to the strong El Niño impact. In particular, we address the following questions: (1) how are cloud radiative and microphysical characteristics influenced by fire particles, (2) does the total amount and/or the diurnal cycle of rainfall change due to fire particles, and (3) what do the simulation results imply, regarding the indirect impacts of ENSO on precipitation via aerosols? Even though the above studies partly gave answers to some of these questions from their simulations, further investigations and discussions, especially on cloud radiative property changes and ENSO impacts, are essential to fully

comprehend ACI in MC. We seek answers to the questions by running a pair of month-long cloud-resolving WRF-CHEM simulations over MC.

## 2   Methods

We utilized the WRF-CHEM model (available at https://www2.mmm.ucar.edu/wrf/users/download/get_source.html) version 3.6.1 for the cloud-resolving simulations. The simulation domain is over MC as Figure 4 shows. The horizontal resolution of this outer domain is 20 km, whose information gets passed to a nested inner domain outlined by the magenta rectangle. This inner domain, which covers the major target region, has a 4-km horizontal resolution and 50 vertical levels. The time steps are 30 s and 6 s for the inner and outer domains, respectively.

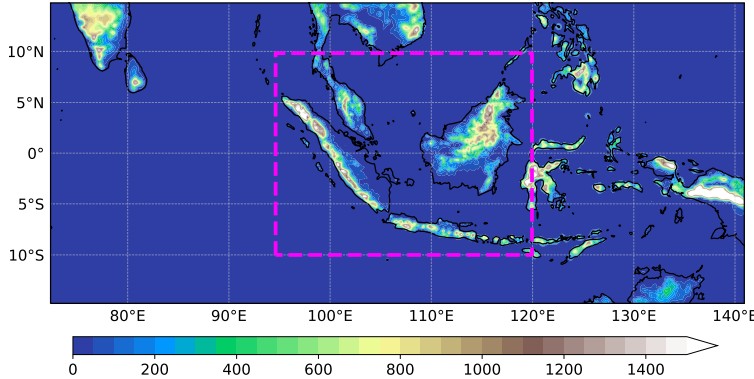

**Figure 4.** Terrain height [m] within the outer simulation domain, outlined by the map. The inner domain is outlined by the magenta rectangle.

Physical and dynamical settings of the simulations are as follows; a simulation with no fire input was initialized at 00UTC on 1 August 2015 by the 1-degree NCEP Final Analysis data (GFS-FNL; NCEP, NWS, NOAA, U.S. DOC, 2000, available at https://www2.mmm.ucar.edu/wrf/users/download/free_data.html) that also provided 6-hourly boundary conditions to the outer domain. This simulation was run for 31 days for spin-up and continued until the end of September 2015, whereas the other simulation with a fire input started from 1 September with the spun-up chemistry field from the no-fire simulation. Therefore, our analysis period is 1-30 September 2015. Microphysical processes were calculated by the two-moment Morrison scheme (Morrison et al., 2009) but its default upper limit on ice number concentration was removed. Longwave and shortwave radiation processes were calculated by the RRTMG scheme (Iacono et al., 2008), land surface processes by the Unified NOAH land surface model (Tewari et al., 2004), and surface layer processes or planetary boundary layer physics by the Mellor-Yamada Nakamishi Niino Level 2.5 scheme (Nakanishi and Niino, 2006, 2009; Olson et al.). Cumulus parameterization, applied solely to the outer domain, was based on the Grell-Freitas Ensemble scheme (Grell and Freitas, 2014). 6-hourly grid-nudging by the Four-Dimensional Data Assimilation (FDDA) was turned on for the outer domain.

Chemistry settings are as follows; emissions were specified by the REanalysis of the TROpospheric chemical composition over the past 40 years (RETRO; Schultz et al., 2008) and the Emission Database for Global Atmospheric Research

(EDGAR; http://edgar.jrc.ec.europa.eu) via Prep-chem-source (Freitas et al., 2011) version 1.5. (available at ftp://aftp.fsl.noaa. gov/divisions/taq/global_emissions/), except for black carbon, organic carbon, CO, $NH_3$, $SO_2$, $NO_x$ (split into 75% NO and 25% $NO_2$), PM2.5, and PM10 that were replaced by the 0.25°-resolution data from the Regional Emission inventory in ASia (REAS; Kurokawa et al., 2013) version 2.1 (available at https://www.nies.go.jp/REAS/). The month-long spin-up period provided enough time for the concentrations of chemical species to stabilize. As for the calculations of chemical processes, we employed the RADM2 chemical mechanism (Stockwell et al., 1990) with MADE/SORGAM aerosols (Ackermann et al., 1998; Schell et al., 2001). Unfortunately, sea salt emissions needed to be turned off as the appropriate emission option was unavailable. As long as we focus on the comparison between the two simulations, we deem this has no impact on our findings. Photolysis was calculated by the Madronich photolysis scheme (Madronich, 1987) and biogenic emissions by the Guenther scheme (Guenther et al., 1994; Simpson et al., 1995).

Under these physical and chemical settings, two simulations were run and compared, with the aim to clarify the impacts of fire particles on clouds and climate. The one with no fire emission is called NOFIRE and the other with high-resolution fire emissions, based on the Fire INventory from NCAR (FINN; Wiedinmyer et al., 2011, available at https://www.acom.ucar.edu/ Data/fire/) version 1.5, is called FIRE. These fire particles were vertically distributed by the embedded plume-rise model based on Freitas et al. (2007). Other than this fire input, everything else remained identical between the two simulations. It is worth noting here that this choice of fire inventory may have a significant impact on the simulated results; for instance, the Quick Fire Emissions Dataset (QFED; Darmenov and da Silva, 2015) provides a relatively large amount of particle emissions from fires compared to FINN (e.g., Liu et al., 2020; Pan et al., 2020). On the other hand, the recently published version of FINN (version 2.4) may include an improvement to the FINN dataset that leads to its large difference from version 1.5 that this study utilized. While the improvements of fire inventories are still ongoing and their comparisons are beyond the scope of this study, the potential impact of using different inventories needs to be kept in mind.

## 3 Results

In this chapter, the comparison of the simulations with observations (Sect. 3.1), the comparison of the two simulations (Sect. 3.2), and the comparison of the findings to those in two previous studies (Sect. 3.3) are presented.

### 3.1 Comparison with Observations

In order to assess how realistic the simulations are, here we compare simulated fields against observations. Figure 5 shows the maps of accumulated surface precipitation [mm] between 1 September and 30 September 2015, observed by TRMM (Figure 5a) and simulated by the model (Figure 5b-c). Despite the general overestimation of the amounts of precipitation in the simulations, the major characteristic distributions are well captured by the model; for instance, large amounts of rainfall in the west of Sumatra (Region 1, red), the southern part of the South China Sea (Region 2, magenta), and in the northern part of the Borneo island (Region 3, yellow). Our analyses focus on these three regions, individually. When daily precipitation patterns are compared, the simulated patterns match surprisingly well with the TRMM observations. According to the time

series in Figure 5d-f, the simulations generally capture the observed fluctuations of rainfall rates, especially over Region 2. The discrepancies in the absolute values shown in Figure 5a-c may be largely due to the lack of ocean dynamics that can lead to more realistic sea surface temperature (SST) distributions. As can be inferred from the effects of ENSO on the amounts of precipitation over MC, SST has significant impacts on the convective activities in the tropics (e.g., Graham and Barnett, 1987; Woolnough et al., 2000; Tompkins, 2001). Indeed, MC lies over the tropical warm pool where the Earth's highest SST is observed. Sabin et al. (2013), for example, analyzed the observed data of SST and convective activities around the warm pool and found a tight connection between the two, especially between 26°C and 29°C. Estimating SST over MC also faces an additional challenge: the complicatedly distributed while nearly equalized coverages of land and ocean in the area. It hence requires a high-resolution ocean model to accomplish (e.g., Wei et al., 2014). Therefore, the lack of realistic temporal and spatial variations in SST may be one of the reasons why the simulated amounts of rainfall are off, while the spatial distributions and the overall temporal evolution are reasonably well reproduced. Some efforts have been already made to couple WRF/WRF-CHEM with an ocean model as in Warner et al. (2010) and Zhang et al. (2019b). Over MC where the varying sea conditions can strongly influence convective activities, the use of such comprehensive models is more desirable and may lead to more realistic simulations.

As for aerosols, there seems to be an overall underestimation of AOD in our simulations on monthly average. Figure 6 shows the maps of monthly mean AOD at 0.55 $\mu$m observed by MODIS Terra (Figure 6a; Aqua shows a similar result) and simulated by the model (Figure 6b-c), which clearly indicates the underestimation, while the FIRE simulation is closer to the observation as expected. The time series of AODs in Figure 6d-f also show this underestimation in each region. Nevertheless, horizontal distributions of high AOD areas are mostly well captured by the FIRE simulations, when the daily snapshots of AODs are compared. Figure 7 compares AODs observed at AERONET stations within the simulation domain with those simulated at the nearest grid points. The accuracy of the FIRE simulation varies from station to station, while the general underestimation of AODs by the simulation is seen. The temporal evolution of AODs, however, seems to be reasonably well captured (e.g., Figure 7d, e, and g). The scale of the AOD values in this figure also needs to be emphasized, as extremely high observed values (e.g., > 2.0) are particularly not well captured by the FIRE simulation.

These comparisons indicate that our simulations capture the overall distributions of rainfall and aerosols, while the amounts of aerosols are likely underestimated. This can be due to the lack of a few large aerosol particles that contribute significantly to AODs because of their large sizes, and/or the lack of many small particles. That is, it is possible that FINN greatly underestimated particle emissions from biomass burning, especially in such a year with extreme dryness. It is also plausible that the volume-based calculations of aerosol optical properties from the aerosol abundance lead to the underestimation of AODs, even though the number and mass concentrations of aerosols are realistic. Unfortunately, there is no observation of aerosol number/mass concentrations that can be compared to our simulations. Given the potential for the underestimation of aerosol mass/number, however, it needs to be kept in mind that the effects of fire particles in reality may have been even stronger than what we find from the comparison of the FIRE and NOFIRE simulations, presented in the following section.

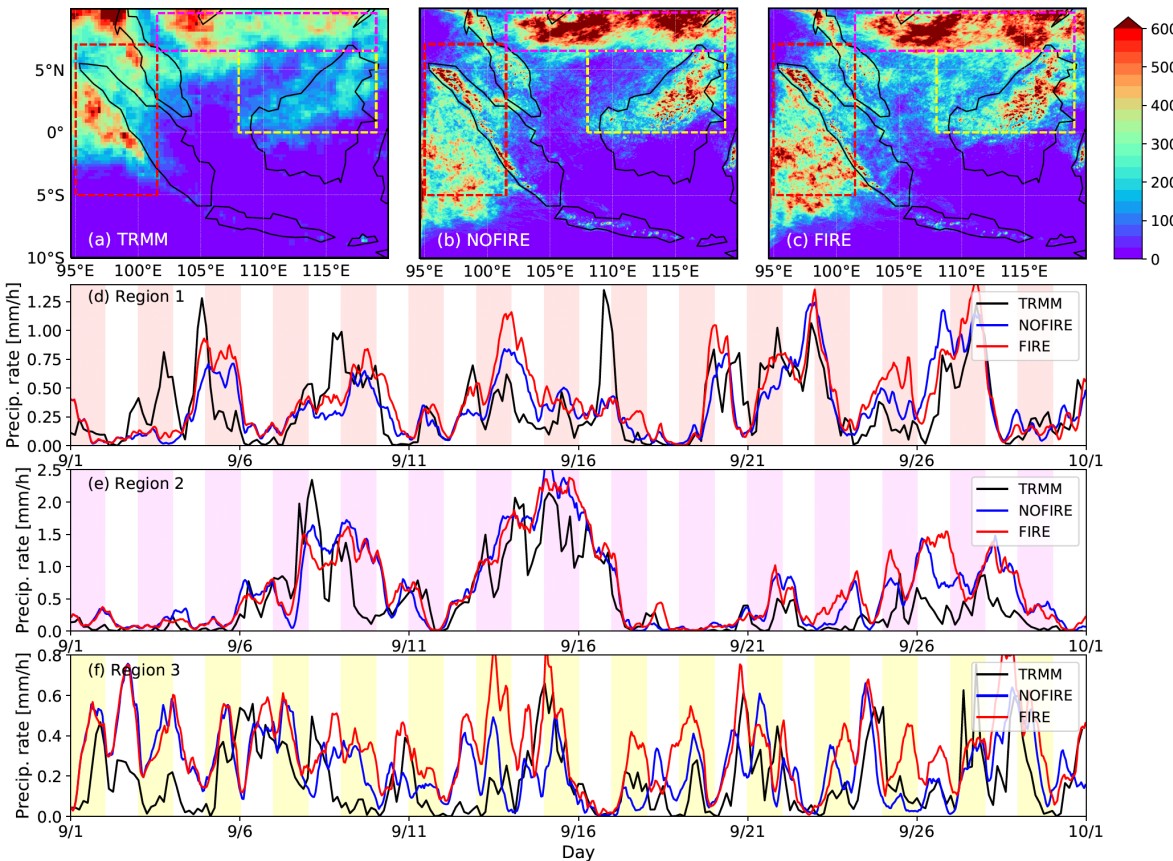

**Figure 5.** Accumulated precipitation [mm] for the month of September in 2015, (a) observed by TRMM and (b) simulated in NOFIRE and (c) FIRE. Red, magenta, and yellow rectangles show the locations of Region 1 (95W-101.5W, 5S-7N), Region 2 (101.5W-119W, 6.5N-9.5N), and Region 3 (108W-119W, 0-6.5N), respectively. Time series of TRMM (black, 3-hourly) and simulated (blue and red, hourly) precipitation rates [mm/h], averaged over each region, are shown for (d) Region 1, (e) Region 2, and (f) Region 3.

### 3.2 FIRE vs. NOFIRE

The inclusion of fire particles led to differences in simulated radiative and microphysical fields. Firstly, their impacts on radiation are discussed. Figure 8a-d shows the mean changes in incoming (ground-level) and outgoing (top-of-the-atmosphere) shortwave radiation under clear- and all-sky conditions, respectively. It is clear from this figure that the inclusion of fire particles reduced the solar radiation reaching the ground by scattering and/or absorbing. Such a radiative difference indeed led to lower temperature near the ground in FIRE, by a degree or so, as shown in Figure 8e. The location of this strongest cooling effect coincides with that of the largest reduction in incoming insolation on the ground, implying their connection. At the top of the atmosphere, outgoing shortwave radiation increases in FIRE particularly over the seas where the surface is dark, due to scattering by fire particles. As a result, albedo increases in the FIRE run (Figure 8f), although this increase is partly

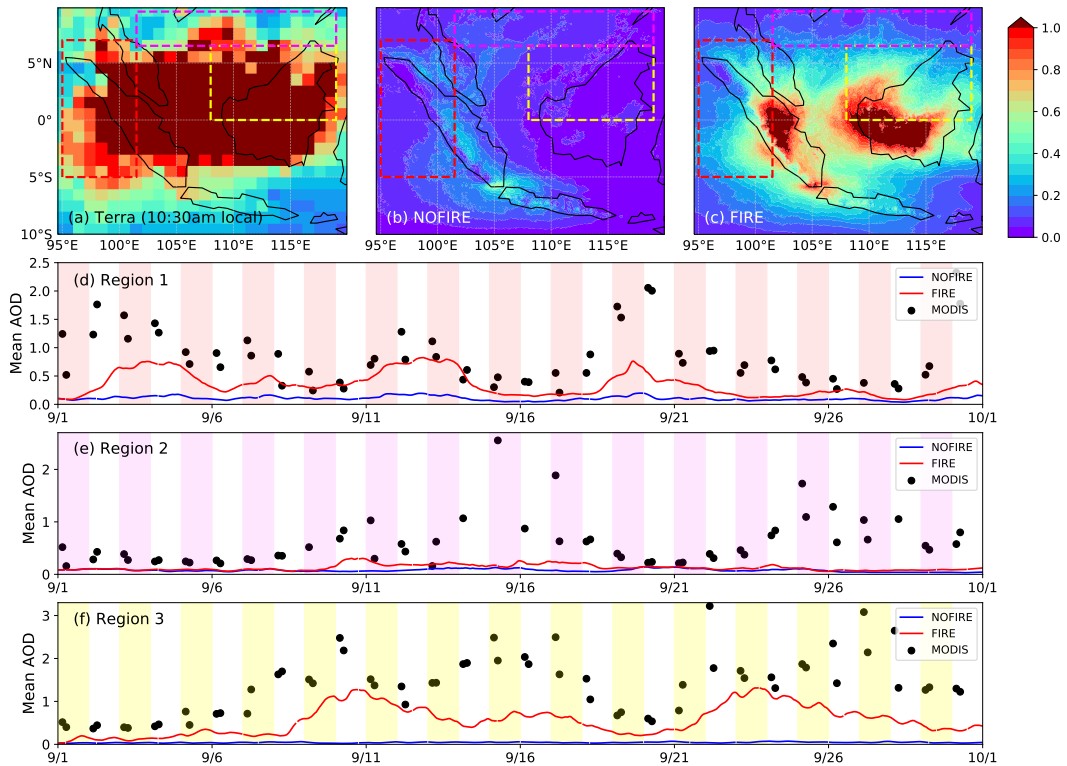

**Figure 6.** Monthly mean AOD at 0.55 $\mu$m (a) observed by MODIS Terra (MOD08_D3) and (b) simulated in NOFIRE and (c) FIRE. AOD from MODIS Aqua (MYD08_D3) shows a similar result. The simulated results in (b,c) are the averages of AOD snapshots at 0320 UTC everyday, which corresponds to 10:20am in Indochina Time. Time series of observed (black dots, twice daily) and simulated (blue and red lines, hourly) AODs, averaged over each region, are shown for (d) Region 1, (e) Region 2, and (f) Region 3. In (d-f), both Terra and Aqua are included. Note that the MODIS data was projected onto the UTC time series in (d-f), assuming that the data was taken at 10:30am/01:30pm in Indochina Time.

due to the increased cloud optical depth (Figure 8g); the aerosol direct and indirect effects both worked to increase the overall reflectivity, even though their timing or causal relationship remains uncertain. The reduction in outgoing longwave radiation

(OLR) in FIRE (Figure 8h) indicates that cloud top heights increased on monthly average. This reduction in OLR suggests that convection was stronger and clouds developed taller in the FIRE run. Although this is contrary to what can be expected from the aerosol radiative effect that reduced the surface temperature and worked to stabilize the atmosphere, we show next that convection became stronger in the FIRE run and increased the amount of rainfall.

    The differences in rainfall between NOFIRE and FIRE, shown in Figure 9a, clearly indicate the enhancement of precip-

itation in the FIRE simulation over the simulation domain. The maximum increase amounts to 645.4 [mm/month] and the domain-mean difference is +25.9 [mm/month]. Based on our analyses, this rainfall difference is a result of a modified chain of microphysical processes, rather than dynamical differences: as mentioned above, aerosol radiative effect seemed to have

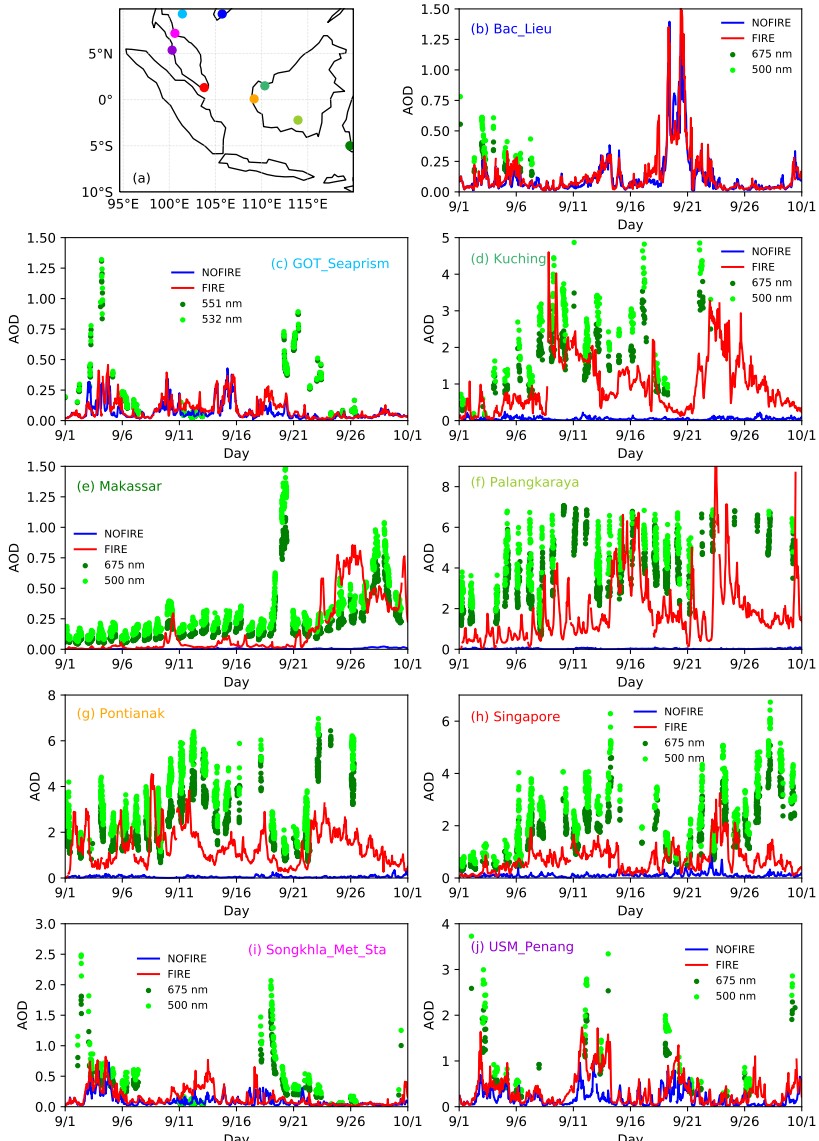

**Figure 7.** (a) Locations of 9 AERONET stations whose AOD data for the month of September 2015 are shown in (b-j) in green (675 nm) and lime (500 nm). For (c), they are AODs at 551 nm (green) and 532 nm (lime) instead. Red and blue lines are estimated AODs at 550 nm from the FIRE and NOFIRE simulations, respectively.

stablized the atmosphere, which has the opposite effect to the invigoration of convection. Therefore, it is fair to state that the enhanced rainfall was triggered by the microphysical effect of fire particles. The rest of this subsection is dedicated to explaining the microphysical mechanisms that made rainfall to increase and clouds to become taller and more reflective in the FIRE simulation as presented above.

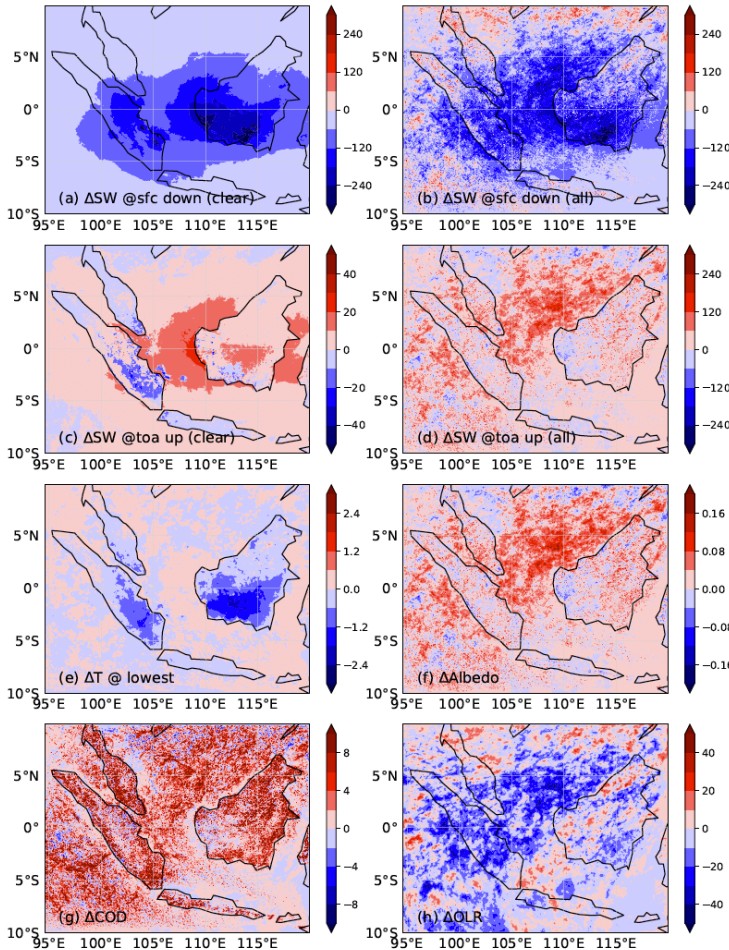

**Figure 8.** Simulated monthly mean differences (FIRE-NOFIRE) in (a) clear- and (b) all-sky downward shortwave radiation on the ground [Wm$^{-2}$], (c) clear- and (d) all-sky upward shortwave radiation at the top of the atmosphere [Wm$^{-2}$], (e) temperature at the lowest model level [K], (f) albedo [unitless], (g) estimated cloud optical depth [unitless], and (h) OLR [Wm$^{-2}$]. These are the averages of differences at 06 UTC (around local midday) everyday in the month of September 2015.

According to Figure 9b-d, the amount of rainfall clearly increases in the FIRE simulation over Region 1 and Region 3, which both include a large portion of land, whereas the precipitation change is equally positive and negative over Region 2 that is over a sea. In order to understand at what time this increase occurs in a day, regional mean differences (FIRE-NOFIRE) in hourly precipitation rate [mm/h] are plotted in Figure 10 (left column), along with their monthly means (magenta). This analysis shows that the increase occurs at 1200-0400 UTC on average, which is from evening to morning in local time. No decrease is seen at other times. To confirm that this increase in nocturnal rainfall is not due to a single passage of a large convective system, the numbers of days with regional mean increase (+1, red, FIRE > NOFIRE) and decrease (-1, blue, FIRE < NOFIRE) are counted every hour in Figure 10 (right column). Combining the plots in Figure 10, it is clear that there are

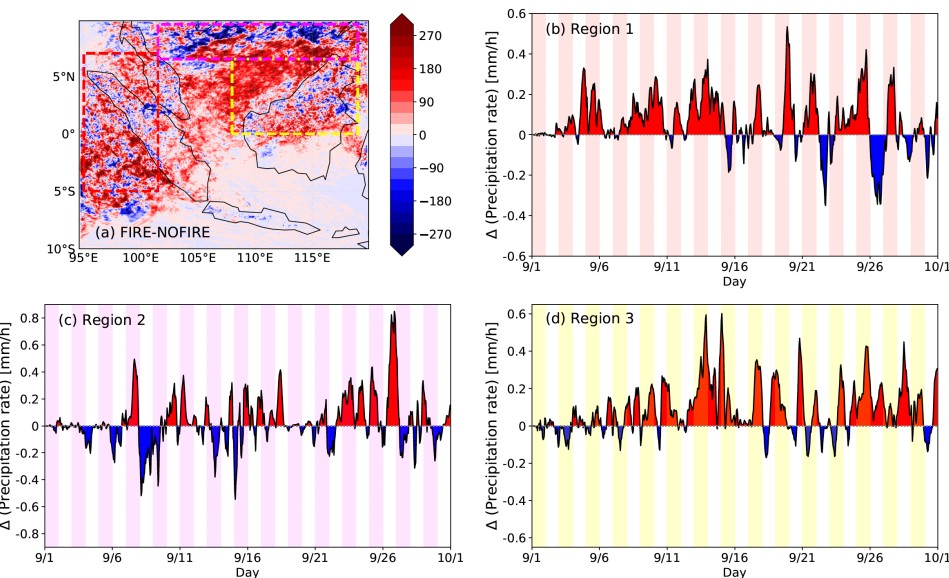

**Figure 9.** (a) Difference (FIRE-NOFIRE) in accumulated precipitation [mm] over the month of September. (b-d) Time series of regional mean precipitation rate differences (FIRE-NOFIRE) in (b) Region 1, (c) Region 2, and (d) Region 3.

many days when precipitation rates increase in FIRE at night (right column), which all contribute to the increase of mean
precipitation rates in 1200-0400 UTC (left column, magenta). As no rainfall reduction is seen for the rest of the day, we
conclude that these precipitation increases over Regions 1 and 3 are an enhancement of nocturnal precipitation rather than a
temporal shifting of a diurnal cycle. When monthly mean mass mixing ratios of hydrometeors are compared between FIRE
and NOFIRE (Figures 11 and A1), an increase in all hydrometeor masses in FIRE is apparent, particularly that of snow and
graupel over Regions 1 and 3. It is also clear that the longitudinal/latitudinal patterns of the rain and snow/graupel masses
correspond very well with each other, implying the significant contribution of melted snow/graupel to rain mass. Furthermore,
the absolute difference values shown in Figures 11 and A1 are on the order of 0.01 [$gm^{-3}$] for both rain and snow/graupel in
all three regions, whereas those for cloud droplets are merely 0.001 [$gm^{-3}$]. Thus, surface rainfall seems to largely stem from
melted snow and graupel. As more aerosol particles exist in the FIRE simulation, the number of smaller droplets increases.
This initiated a chain of altered microphysical processes that led to the increase in snow production, which is essential for
subsequent graupel production. Further analyses on microphysical process rates (Figure A2) show cloud features that are
consistent with both of the hypotheses that (i) more cloud droplets remain in air without raining out and later freeze aloft to
form more ice crystals that are essential for more snow formation in FIRE and (ii) the increased mass and number of cloud
droplets in the FIRE simulation set a more favorable condition for efficient snow and graupel production in clouds, such as
through droplet accretion by snow (Figure A2g-i). Both of these paths may have concurrently played a role in producing more
snow in the FIRE simulation. Once snow mass increases, graupel mass also increases as they form on snow by riming. As
for the invigoration of convection signified by the increased rainfall and cloud top height, our analysis has revealed that the

increased latent heat release is predominantly through increased condensation rather than increased freezing; Figure A3 shows the estimated rates of maximum latent heat release following droplet activation and freezing. According to these estimates,

convection was likely invigorated more by increased condensation and less so by increased freezing. This result agrees with what was shown by Fan et al. (2018) and Lebo (2018). As a result of the enhanced condensation, the maximum supersaturation is lowered inside convection in FIRE (Figure A4). It is also likely that, once convection gets invigorated, stronger downdrafts can in turn induce stronger convection, creating a positive feedback. This may have played a role in the invigorated convections in our FIRE simulation.

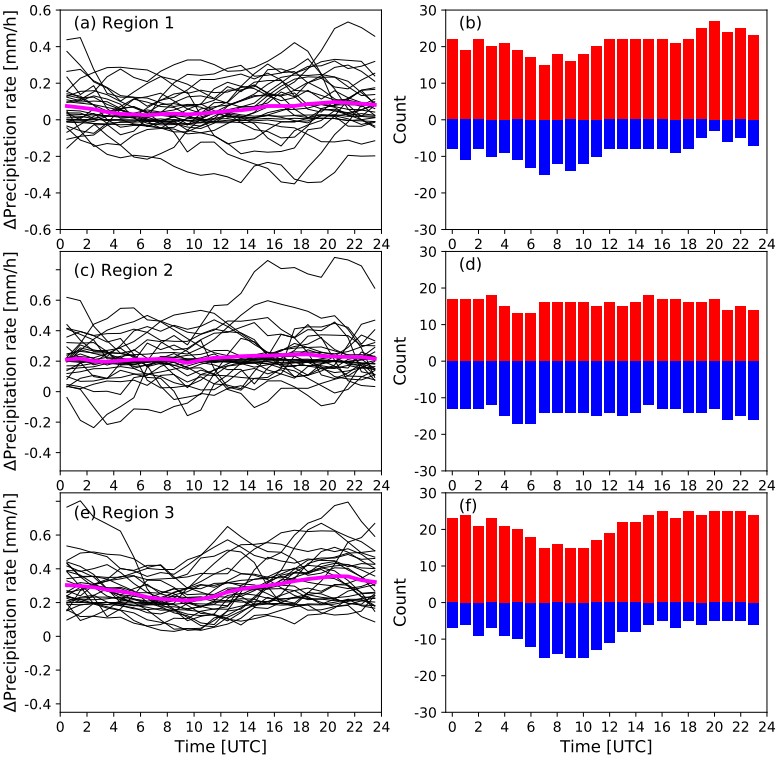

**Figure 10.** (a,c,e) FIRE-NOFIRE differences in hourly precipitation rate [mmh$^{-1}$] each day in September (black) and their monthly average (magenta). (b,d,f) Raw counts of increased (+1, red) or decreased (-1, blue) hourly rainfall rates (FIRE-NOFIRE). All are averages over (a,b) Region 1, (c,d) Region 2, and (e,f) Region 3.

Areas with increased cloud optical depth (Figure 8g) overlap with areas of increased droplet number concentrations (over land) and areas of increased ice mass (over sea), based on Figure 12. This suggests the direct impacts of fire particles on cloud droplets over land (i.e., source region) and their propagated impacts over seas on cloud reflectivity via ice. As for cloud top heights, the areas of decreased OLR (Figure 8h) correspond well with the areas of increased ice (Figure 12d-f), mostly over seas. This increased mass of ice crystals is likely due to the hypothesis (i) stated above, which applies to convective

clouds that can eventually dissipate into more stratiform anvil clouds and drift in the atmosphere for an extended period of

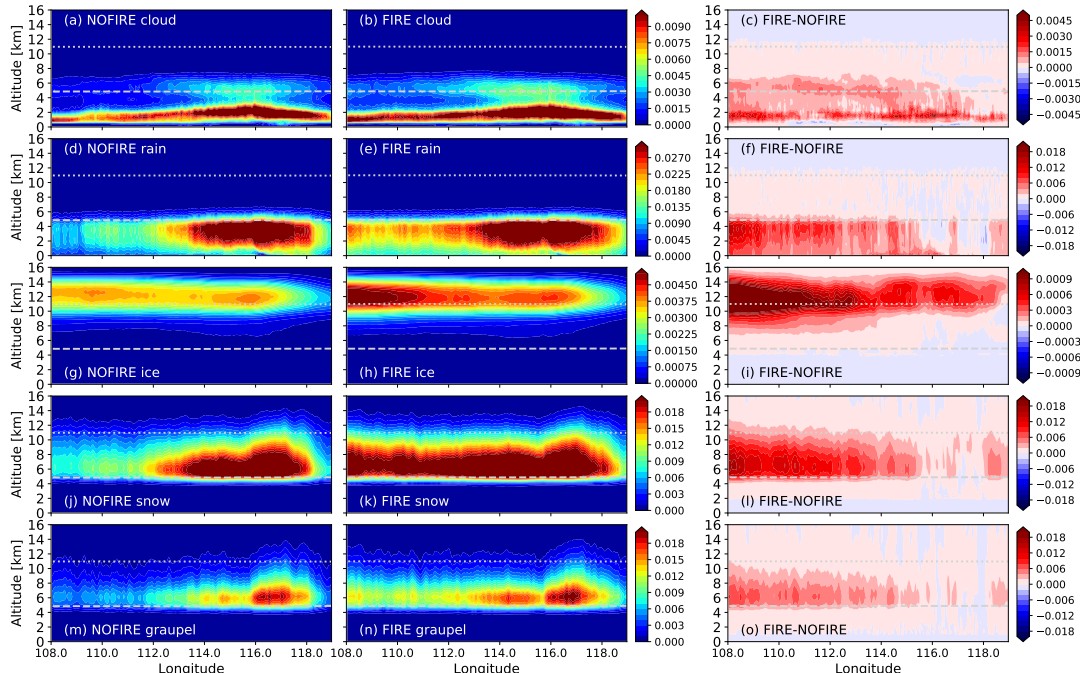

**Figure 11.** Monthly and meridional mean mass concentrations [gm$^{-3}$] of (a,b) liquid cloud, (d,e) rain, (g,h) ice, (j,k) snow, and (m,n) graupel in (a,d,g,j,m) NOFIRE and (b,e,h,k,n) FIRE over Region 3. These are averages inside grid boxes where each hydrometeor mass > 0. Respective differences (FIRE-NOFIRE) are shown in the rightmost column. The dashed and dotted lines are temporally and meridionally averaged 0°C and -40°C isotherms, respectively. See Figure A1 for equivalent difference figures for Regions 1 and 2.

time. Such aerosol-induced changes in stratiform anvil clouds, namely their extended lifetime and heightened cloud top, have been reported in previous studies (e.g., Fan et al., 2013), even though these changes can be independent of the invigoration of convection. The increased ice mass indicates increased latent heat release in FIRE through freezing and riming, which is consistent with the higher cloud tops. Thus, an increase of fire particles triggered the changes in microphysical process rates

that ultimately modified the cloud radiative properties.

These microphysically-driven changes of cloud properties have some implications on climate. The invigoration of convective clouds produces more rainfall and hence facilitates the energy exchange between the surface and the upper atmosphere. Interestingly, the increase of fire particles in the atmosphere is partly dependent on the amount of precipitation; the year of 2015 was particularly dry and had an exceptionally high AODs (Figure 2) due to the lack of rainfall (Figure 1). Although the amount

of increased rainfall in FIRE is not as much as interannual differences, our simulations showed the effect of fire particles to slightly compensate for the lack of rainfall in the year of 2015.

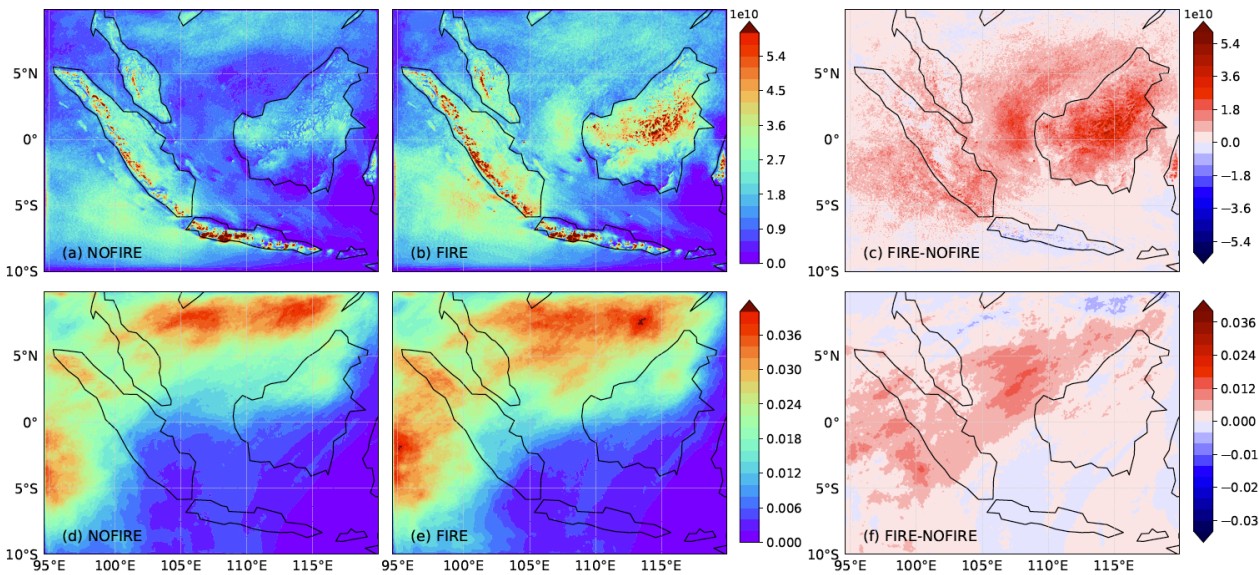

**Figure 12.** Monthly mean column-integrated (a,b) cloud droplet number [m$^{-2}$] and (d,e) ice mass [kgm$^{-2}$] in (a,d) NOFIRE and (b,e) FIRE. Differences are shown in (c,f).

### 3.3 Comparison with Previous Studies

Here we compare the results in this study with those from HD18 and LW20. The objective of this subsection is to clarify the differences in the overall impacts of ACI among the three studies, particularly in the surface rainfall changes. As simulation settings differ among the three, only the signs of the rainfall change are concerned rather than its magnitude or physical mechanism. The simulation period was mostly September in HD18 and 4 months from June to September in LW20. The month of September, therefore, is the common period of interest among these studies and the current work. The ENSO phase, however, differs among the three: very strong El Niño in 2015 (this study), moderate El Niño in 2009 (HD18), and weak La Niña in 2008 (LW20), according to the 3-month running mean Oceanic Niño Index (https://origin.cpc.ncep.noaa.gov/products/analysis_monitoring/ensostuff/ONI_v5.php). Although the simulation resolutions are more or less the same cloud-resolving scale for the three (i.e., 4 km or 5 km), the simulation domain contains both the Indochina and MC in LW20, only MC in this study, and only the Borneo island in HD18. With these differences in mind, the findings of rainfall changes due to fires over the Borneo island are compared in this subsection.

Our results show the increase in nocturnal rainfall over Borneo (i.e., Region 3) due to the intensification of convective clouds through microphysical processes. The direct and semi-direct effects of aerosols were small. In HD18, however, the inclusion of fires reduced the rainfall from late afternoon to evening (see their Figure 5) due to radiative (stabilization of the atmosphere) and microphysical (locally vary) reasons. As for LW20, rainfall slightly increased during daytime but decreased during nighttime when fires were included (see their Figure 10). LW20 concluded that it was likely due to the semi-direct effect of aerosols, which

reinforced sea breeze during the day but weakened land breeze at night. Over the same region and same month, these studies have varying results for the fire-induced rainfall change and its mechanism. We attribute these differences to (a) microphysical paramterizations and their sensitivities to aerosol perturbations, (b) settings of aerosol properties, such as absorptivity and size distributions, that would affect the sign and/or magnitude of aerosol radiative effects, and (c) the interannual variability of dominant regional weather pattern that prevails over SEA, likely varying with the ENSO phase. As for (a) and partly for (b), both LW20 and this study used the Morrison scheme (Morrison et al., 2009), but the removal of the default upper limit on ice number concentrations in this study may have increased the sensitivity of clouds to aerosol perturbations. HD18 utilized the two-moment scheme by Thompson and Eidhammer (2014) that separates aerosols into water-friendly and ice-friendly and activates a fraction of water-friendly aerosols based on a look-up table. These differences in the calculations of microphysical processes must have a considerable influence on the simulated results of the fire effects. The additional factor of (c) may further complicate the interpretation of the differences among simulations. Therefore, comparisons of simulations with a consistent microphysics scheme or the same ENSO phase are required for fully clarifying the role of fire particles on clouds or the effects of ENSO phases on ACI over the region.

## 4   Conclusions

We have used two cloud-resolving WRF-CHEM simulations to reveal the impacts of fire particles on cloud microphysics and radiation over MC for the month of September in 2015, when extremely high AODs were observed. Our month-long FIRE simulation with fire particles showed more reflective and taller clouds than those in the NOFIRE simulation. The amount of precipitation was also larger in the FIRE simulation. All of these features suggest the invigoration of convection by fire particles. Based on our further analyses, the increased mass of snow seemed to be particularly responsible for the increased rainfall, whereas the changes in cloud top height and reflectivity stemmed mainly from increased ice crystals that are more reflective and longer-lived than snow. The changes in microphysical process rates were all initiated by a simple increase in aerosol number concentrations, which in turn triggered a chain of modified microphysical processes such as increased freezing of smaller ice crystals aloft and thermodynamic responses. Although the magnitude of the differences between FIRE and NOFIRE is not comparable to interannual differences, we conclude that the intensification of convection by fire particles acted to partly compensate for the lack of rainfall for the month of September 2015. These findings answer the three scientific questions posed for this study in Introduction.

It is of a profound interest to understand the interannual variability of aerosol effects potentially influenced by ENSO, which is commonly believed to be a major driver for convective activities and also a critical factor behind biomass burning in SEA. To explore this issue, we have compared our simulations with two previous studies, perhaps the only other cloud-resolving, month-long simulations available hitherto over the region but for years in different ENSO phases. Convective systems in our simulations displayed the invigoration effect by fire particles that the two studies did not show. Nevertheless, many questions still remain unanswered. For instance, do smaller amounts of aerosols in other years simply exert the weaker invigoration effect that was presented in this study? Or even small amounts of aerosols have an equivalently strong invigoration effect if

the background condition is always pristine, unlike the aerosol-loaded year of 2015? Does the aerosol semi-direct effect play a stronger role in other years, or depending on the aerosol settings in simulations? Are aerosol effects completely different under different weather regimes? Although it was out of the scope of this paper, recent studies such as Zhang et al. (2019a)

290 have shown a potential importance of heat effects of fires on convective clouds; they found strengthening of convection by the heat effects and therefore significant changes in subsequent cloud properties in their simulations. In the region of our interest, how much the heat effects of fires exert the invigoration of convection could definitely be examined in the future. Furthermore, do our simulation results depend on the fire inventory used for the simulations? The answers to all of these questions will be sought in our future work.

**Appendix A: Supplementary Figures**

This appendix provides additional figures that support the contents of the paper.

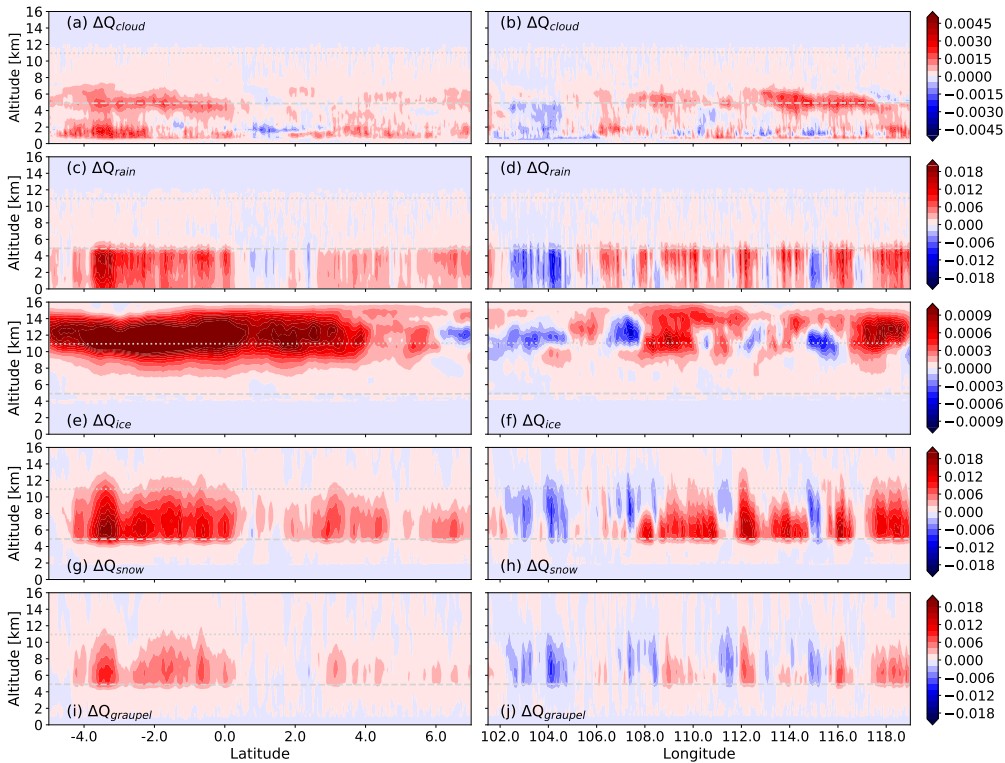

**Figure A1.** Same as (c,f,i,l,o) in Figure 11 but for (left column) Region 1 and (right column) Region 2.

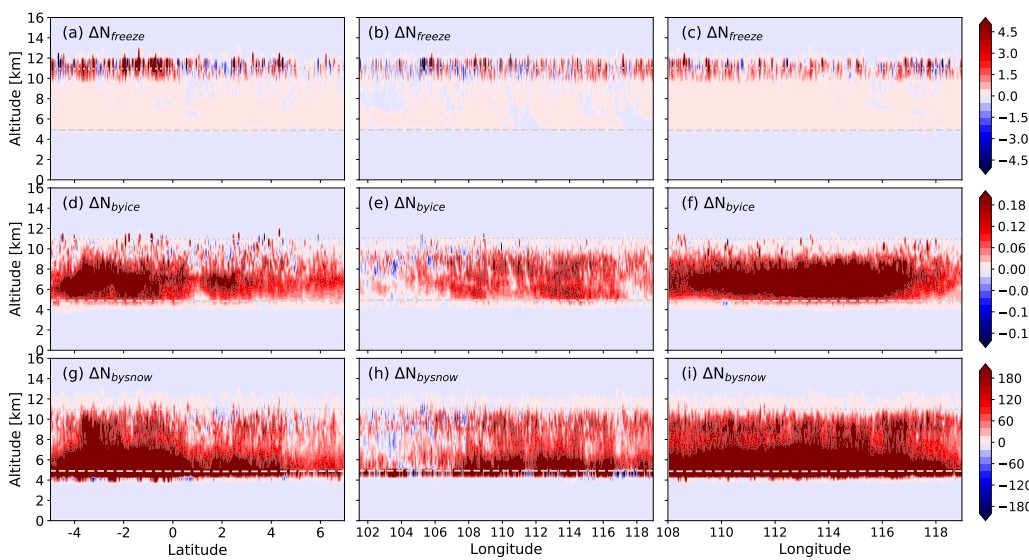

**Figure A2.** Differences (FIRE-NOFIRE) in rates [#kg$^{-1}$s$^{-1}$] of (a-c) cloud droplet freezing, (d-f) droplet accretion by ice, and (g-i) droplet accretion by snow in (left column) Region 1, (middle column) Region 2, and (right column) Region 3. These are averages inside grid boxes where rates > 0.

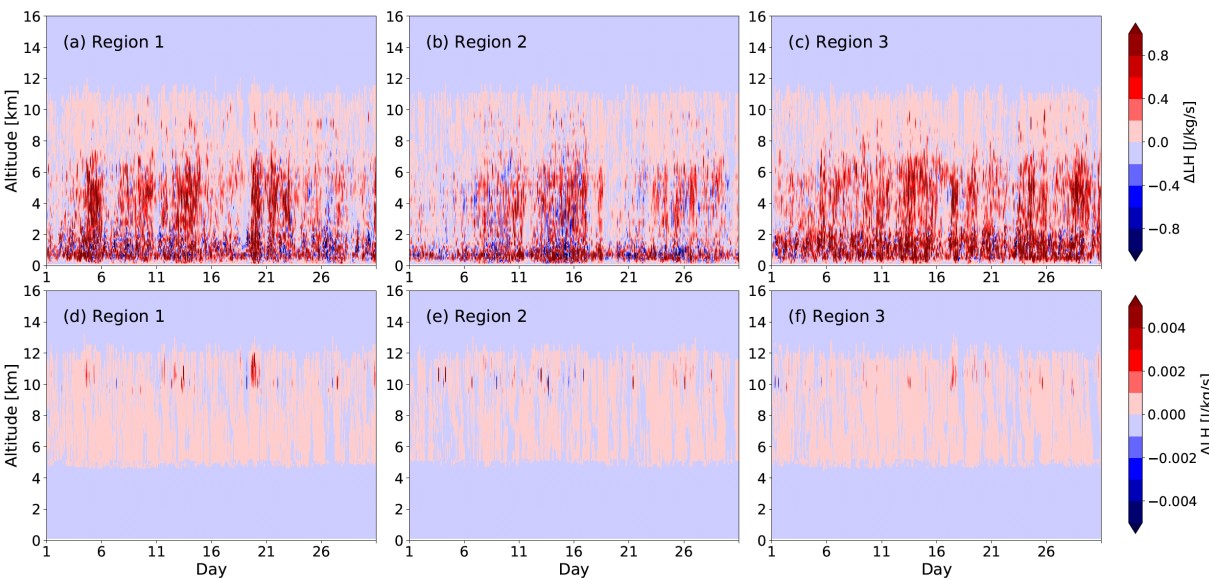

**Figure A3.** Differences (FIRE-NOFIRE) in the estimated rate of maximum latent heat release $[\mathrm{Jkg}^{-1}\mathrm{s}^{-1}]$ following (top) droplet activation and (bottom) droplet freezing in (a,d) Region 1, (b,e) Region 2, and (c,f) Region 3. These were estimated from newly activated droplet number concentration $[\#\mathrm{kg}^{-1}]$, time step [s], droplet freezing rate $[\#\mathrm{kg}^{-1}\mathrm{s}^{-1}]$, and cloud droplet and ice effective radii $r_c$ and $r_i$. Since newly formed droplets and ice crystals are typically smaller than $r_c$ and $r_i$, respectively, these are the maximum estimates. Note the difference in the scale of the color bars.

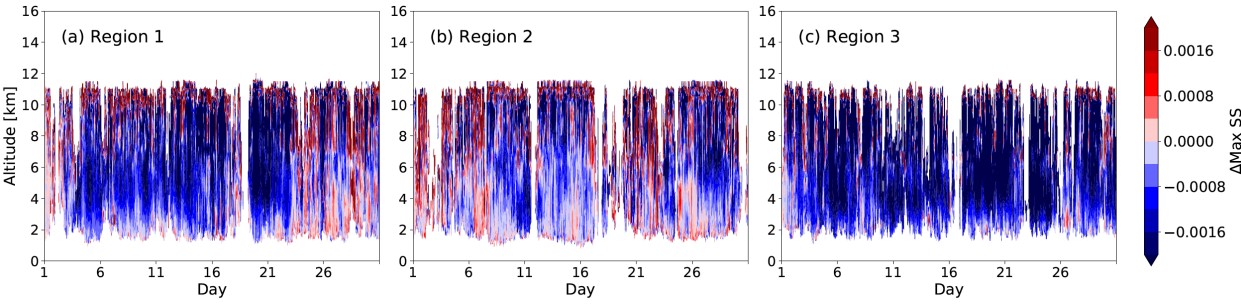

**Figure A4.** Differences (FIRE-NOFIRE) in maximum supersaturation $S_{max}$ averaged within each region, only sampled where updraft $\geq 5$ $\mathrm{ms}^{-1}$ and $S_{max} > 0$.

*Code and data availability.* The source codes of the model used in the study, the Weather Research and Forecasting model coupled with Chemistry (WRF-CHEM), are publicly available at https://www2.mmm.ucar.edu/wrf/users/download/get_source.html. All the other data used for running our simulations are also made available to the public by various responsible institutions. Readers interested in the specific

modifications made to the WRF-CHEM source code for this study can contact the corresponding author.

*Author contributions.* A. Takeishi and C. Wang conceptualized the research theme and designed the simulations. A. Takeishi ran the simulations, analyzed and visualized the results, and wrote the original draft of the manuscript. C. Wang supervised the overall work and contributed to the preparation of the submitted manuscript.

*Competing interests.* The authors declare that they have no conflict of interest.

*Acknowledgements.* This work is a part of the *Make Our Planet Great Again* project and funded by L'Agence National de la Recherche (ANR) of France under "Programme d'Investissements d'Avenir (ANR-18-MOPGA-003 EUROACE)". We appreciate the computational support provided by the Institute for Development and Resources in Intensive Scientific Computing (IDRIS) and the Grand Équipement National de Calcul Intensif (GENCI) for the numerical simulations in this work. We thank all the corresponding institutions for making their data products available for this study. We also thank the two anonymous reviewers for their insightful comments and suggestions that allowed

310 us to significantly improve the paper.

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
