# Peer review of "Radiative and microphysical responses of clouds to an anomalous increase in fire particles over the Maritime Continent in 2015"

_Atmospheric Chemistry and Physics, 2021_

## Referee Comment (RC1)

Comments on the manuscript entitled "Radiative and microphysical responses of clouds to an anomalous increase in fire particles over the Maritime Continent in 2015" by Takeishi and Wang.

General comments:

The authors have investigated the impact of fire emissions on the radiation and microphysics field around Kalimantan Island. The scope of this article is quite essential, and the scientific significance of this article might be considerable, as the interaction between aerosols and clouds is not investigated over the Maritime Continent. However, I regret to say that the quality of the scientific approach and presentation seem insufficient to be published in this journal.

1) FINNv1.5 is unsuitable for significant fire events in the equatorial region (cf. Liu et al. 2020, https://doi.org/10.1016/j.rse.2019.111557).

   The below figure denotes monthly PM2.5 emission accumulated over the southern part of Kalimantan Island [109-118, 3S-3N, same as the rectangle of "Kalimantan" in Figure 2]. FINNv1.5 uses MODIS active fires and has the advantage of detecting small fires, but it does not include peat fires. That means the interannual variation of FINNv1.5 is relatively small compared to that of other fire emission inventories.

[Figure]

2) A detailed analysis has not been conducted to reveal the cause of "*The simulated response of clouds to fire particles in our simulations clearly differs from what was presented by two previous studies that modeled aerosol-cloud interaction in years with different phases of El Niño–Southern Oscillation (ENSO)*" (lines 13-15).

In general, the authors have conducted "*as is*" the model was released. Then, compared with the previous research, and concluded as "*it differs.*" HD18 uses Thompson and Eidhammer (2014) as the microphysics parameterization. It means they do not consider the indirect effect of aerosols because "*the user needs to select a double microphysics scheme; either Lin et al. or the Morrison microphysics schemes are the current possible choices*" (Section 4.3.3 of WRF/Chem Users' Guide; https://ruc.noaa.gov/wrf/wrf-chem/Users_guide.pdf). Also, LW20 implemented special treatment for peatland fires as they wrote as "*The plume rise algorithm in WRF-Chem, specifically modified to improve the representation of tropical peat fire, was described in Lee et al. (2017)*" (Section 2.1 of LW20). The authors should show how these differences affect the results. I also note here that it might be helpful to conduct an additional investigation for the reliability of the result of 4-km grid spacing, as "*The characteristics of the simulated diurnal precipitation cycle changed at a grid spacing of around $2-3\,km$*" (Yashiro et al., 2016, SOLA, doi:10.2151/sola.2016-053).

Detailed comments:

1) Section 2., lines 80-81: The authors should show the version of WRF/Chem applied for this study. It seems the authors have used a 5:1 ratio for the nesting; there might be a problem until version 4.2 as described as "*All of the odd nest ratios (excluding 3:1) had an incorrect feedback. The indexing was off by (ratio/2 -1). For example, a 5:1 ratio would be off by a single fine grid point. Even ratio feedback was not impacted*" (cf. 1st para. of "Others"

section of version 4.2 in https://github.com/wrf-model/WRF/releases).

2) Section 2, last para. (Lines 109-112): The authors should show the version of the FINN emission inventory. Liu et al. (2020) had investigated PM2.5 exposure in Singapore using five emission inventories for 2015. They found that "*FINNv1.5 consistently underestimates smoke PM2.5 in high fire intensity years and most poorly captures the temporal variability of observed smoke PM2.5*" (Section 3.3.1 of Liu et al., 2020). Therefore, I doubt using FINNv1.5 for this kind of research as peat fires are not included in FINNv1.5 (cf. Table 1 of Liu et al. 2020), although peatland is distributed in the southern part of Kalimantan Island. If there is a reasonable reason for using FINNv1.5, the authors should explain it in the revised manuscript. As fire emission is the key to this research, I would like to ask the authors to give additional information in the revised manuscript for 1) the treatment of plume rise calculation (biomass_burn_opt), and 2) diurnal variation. For example, in HD18, "*Biomass burning emissions are based on the Fire Inventory from NCAR (FINN, Wiedinmyer, et al., 2011) and were distributed using the online plume rise model (Grell et al., 2011). These emissions have a diurnal cycle, with maximum values at 15 local standard time (LST), which is important to consider to obtain a realistic diurnal variation of the aerosol optical thickness (Hodzic et al., 2007; Wang et al., 2006)*" (Section 2.1 of HD18). I also note here that if you have used 'grid_finn_fire_emis_v2020' (https://www.acom.ucar.edu/Data/fire/) for the conversion of FINN emission, the maximum value can be seen at 13LST as the default setting (given by "lt_fac" in fire_file.f90). I am not sure there is a diurnal variation at significant fire burning. Still, the authors should explain the diurnal variation used in this study, and I think the diurnal variation should be validated by observed diurnal cycles of AOD (or AOT of HIMAWARI-8).

3) Section 3.1, lines 129-131: I cannot understand why the authors insist "*AOD measurement data from the nearby AERONET stations is unavailable.*" It

seems several AERONET stations had AOD data in September 2015 ([https://aeronet.gsfc.nasa.gov](https://aeronet.gsfc.nasa.gov), I have accessed on 19 October 2021). For the conversion of 500nm AERONET AOD with 550nm AOD, equation 1 of Jiang et al. (2019; https://doi.org/10.3390/rs11091011) could be useful.

[Figure]

Figure for the location of AERONET stations near the target region, which has AOD data in September 2015.

Technical corrections:

1) Page 8, Figure 6: It seems the text "*AOD from MODIS Aqua (MYD08_D3) shows a similar result*" is not suitable for the figure caption. If there is no significant difference between Aqua and Terra, the authors should mention it in the main text.

---

## Author Comment (AC1)

Response to Anonymous Referee #1

**We thank the reviewer for the comments and suggestions. Please find our responses (in bold blue) to the comments and questions below.**

General comments:
The authors have investigated the impact of fire emissions on the radiation and microphysics field around Kalimantan Island. The scope of this article is quite essential, and the scientific significance of this article might be considerable, as the interaction between aerosols and clouds is not investigated over the Maritime Continent. However, I regret to say that the quality of the scientific approach and presentation seem insufficient to be published in this journal.

1) FINNv1.5 is unsuitable for significant fire events in the equatorial region (cf. Liu et al. 2020, https://doi.org/10.1016/j.rse.2019.111557). The below figure denotes monthly PM2.5 emission accumulated over the southern part of Kalimantan Island [109-118, 3S-3N, same as the rectangle of "Kalimantan" in Figure 2]. FINNv1.5 uses MODIS active fires and has the advantage of detecting small fires, but it does not include peat fires. That means the interannual variation of FINNv1.5 is relatively small compared to that of other fire emission inventories.

[Figure]

**Response: We appreciate the additional figure provided by the reviewer, which suggests that the interannual variability is more pronounced in FINN version 2.4 than in version 1.5 that we used for our simulation. It is worth indicating, however, that the version 2.4 seems to have become available only after July 2021, or just about one month before the initial submission of this paper, not to mention that many months were needed for data preparation, model configuration, performing the expensive simulations, and analyzing the results in prior to the paper submission. This was the reason why we used version 1.5. Although new datasets will likely be produced one after another, we think that we can still conduct quality research with existing datasets. Nevertheless, the reviewer's point on the difference between the two versions of FINN has been well received here and we are willing to run a new FIRE simulation with FINN version 2.4 in the near future to see if the new FINN data produces a**

very different result from the original FIRE run (i.e., version 1.5). In this case, we would utilize FINN version 2.4 (MODIS) rather than 2.4 (MODIS + VIIRS) due to the data availability for the year of 2009 that is essential for our interannual comparison study (* different paper).

In addition, we would like to indicate that the emitted primary aerosol mass is not necessarily linearly translated into the mass and number concentrations of aerosols in the atmosphere. For instance, the plume rise model embedded in WRF-CHEM and also how aerosols are horizontally dispersed by winds are both important factors that determine the total aerosol abundance in the atmosphere. Moreover, large aerosol particles typically dominate the total volume and often sediment quickly on the ground, which means that the differences in number concentrations between FINN version 1.5 and 2.4 may be rather small.

We have compared FINN (v1.5) with another fire inventory GFED before the initial submission of the paper. The figure below shows the mean daily emission rate of PM2.5 in FINN (left) and GFED (right) for the month of September in 2015. Although they are gridded differently (FINN: $0.1° \times 0.1°$, GFED: $0.25° \times 0.25°$), we found a substantial difference between them (i.e., FINN v1.5 > GFED), which differs from what was shown in the figure provided by the reviewer above (i.e., GFED > FINN v1.5). It would be helpful if the reviewer could provide more details on the data sources and their availability, so that we could further examine the quantitative differences between FINN v1.5 and GFED.

[Figure]

**Figure:** Mean daily emission rates of PM2.5 in FINN v1.5 (left, $0.1° \times 0.1°$) and GFED (right, $0.25° \times 0.25°$), averaged over the month of September 2015.

2) A detailed analysis has not been conducted to reveal the cause of "The simulated response of clouds to fire particles in our simulations clearly differs from what was presented by two previous studies that modeled aerosol-cloud interaction in years with different phases of El Niño–Southern Oscillation (ENSO)" (lines 13-15).

In general, the authors have conducted "as is" the model was released. Then, compared with the previous research, and concluded as "it differs." HD18 uses Thompson and Eidhammer (2014) as the microphysics parameterization. It means they do not consider the indirect effect of aerosols because "the user needs to select a double microphysics scheme; either Lin et al. or the Morrison microphysics schemes are the current possible choices" (Section 4.3.3 of WRF/Chem Users' Guide; https://ruc.noaa.gov/wrf/wrf-chem/Users_guide.pdf). Also, LW20 implemented special treatment for peatland fires as they wrote as "The plume rise algorithm in WRF-Chem, specifically modified to improve the representation of tropical peat fire, was described in Lee et al. (2017)" (Section 2.1 of LW20). The authors should show how these differences affect the results. I also note here that it might be helpful to conduct an additional investigation for the reliability of the result of 4-km grid spacing, as "The characteristics of the simulated diurnal precipitation cycle changed at a grid spacing of around 2−3 km" (Yashiro et al., 2016, SOLA, doi:10.2151/sola.2016-053).

**Response: The subsection 3.3, commented by the reviewer, serves a purpose of presenting the findings from different studies over the same region and assessing whether there is any consensus on the impacts of biomass burning particles on clouds in the region (i.e., there isn't, as our study showed an increase in precipitation with fire particles, HD18 showed a decrease instead, and LW20 rather found a shift in the diurnal cycle of precipitation when fire particles were included). This objective will be made even clearer at the beginning of the subsection in the revised manuscript. Since all the referred studies used different model/aerosol settings and simulated different years with different ENSO phases, it is not feasible to identify the exact reason why the findings are different.**

**Regarding the Thompson and Eidhammer scheme: The microphysics scheme used in HD18 and developed by Thompson and Eidhammer (2014) is so-called the "aerosol-aware Thompson scheme" and has the advantage of including the effects of aerosol particles on clouds without explicitly simulating many kinds of aerosol particles. Unlike the two schemes (Lin & Morrison) that the reviewer mentioned and are fully coupled to aerosol modules (e.g., the Köhler theory), this scheme splits existing aerosols into water- and ice-friendly, uses a look-up table to activate a fraction of water-friendly particles, and determines the droplet numbers. Therefore, the aerosol indirect effects were still included in this study, only in a different way.**

**Regarding the modified plume-rise algorithm in Lee et al. (2017): As mentioned above, there are several differences in the simulation configurations between our study and LW20, which makes it not feasible for us to isolate the impact of a single different factor. The objective of this paper is to compare the twin simulations, NOFIRE and FIRE, to reveal the impact of fire particles on clouds. Again, the subsection 3.3 simply highlights the different conclusions on the aerosol-cloud interaction over the same region in the three studies, without clarifying the reasons why they are different.**

**As for the horizontal resolution, what resolution is "high enough" depends on the actual modeling task. A change of resolution would definitely bring some differences in the results, though they might not be important for some research tasks. We deem that 4-km resolution is generally considered to be within the "cloud-resolving" scale as in HD18 and**

**LW20, and in other earlier studies. In addition, considering the large domain size and the month-long simulation period, the present resolution is close to the finest resolution that we can afford, computationally.**

Detailed comments:
1) Section 2., lines 80-81: The authors should show the version of WRF/Chem applied for this study. It seems the authors have used a 5:1 ratio for the nesting; there might be a problem until version 4.2 as described as "All of the odd nest ratios (excluding 3:1) had an incorrect feedback. The indexing was off by (ratio/2 -1). For example, a 5:1 ratio would be off by a single fine grid point. Even ratio feedback was not impacted" (cf. 1st para. of "Others" section of version 4.2 in https://github.com/wrf-model/WRF/releases).
**Response: We will state the version of the WRF-CHEM model in this study, which is 3.6.1, in the revised manuscript. Thank you for your suggestion.**

**We deem that the ratio of 5:1 is not an uncommon parent-to-nested ratio and has been used in similar studies. It is not entirely clear to us what/which incorrect feedback (e.g., relevant to aerosol-cloud interaction?) was being referred to, and indeed, we would greatly appreciate it if the reviewer could provide some relevant publication on this matter. Even if the indexing was slightly off, we do not believe that it has an impact significant enough to invalidate our scientific findings from our simulations.**

2) Section 2, last para. (Lines 109-112): The authors should show the version of the FINN emission inventory. Liu et al. (2020) had investigated PM2.5 exposure in Singapore using five emission inventories for 2015. They found that "FINNv1.5 consistently underestimates smoke PM2.5 in high fire intensity years and most poorly captures the temporal variability of observed smoke PM2.5" (Section 3.3.1 of Liu et al., 2020). Therefore, I doubt using FINNv1.5 for this kind of research as peat fires are not included in FINNv1.5 (cf. Table 1 of Liu et al. 2020), although peatland is distributed in the southern part of Kalimantan Island. If there is a reasonable reason for using FINNv1.5, the authors should explain it in the revised manuscript. As fire emission is the key to this research, I would like to ask the authors to give additional information in the revised manuscript for 1) the treatment of plume rise calculation (biomass_burn_opt), and 2) diurnal variation. For example, in HD18, "Biomass burning emissions are based on the Fire Inventory from NCAR (FINN, Wiedinmyer, et al., 2011) and were distributed using the online plume rise model (Grell et al., 2011). These emissions have a diurnal cycle, with maximum values at 15 local standard time (LST), which is important to consider to obtain a realistic diurnal variation of the aerosol optical thickness (Hodzic et al., 2007; Wang et al., 2006)" (Section 2.1 of HD18). I also note here that if you have used 'grid_finn_fire_emis_v2020' (https://www.acom.ucar.edu/Data/fire/) for the conversion of FINN emission, the maximum value can be seen at 13LST as the default setting (given by "lt_fac" in fire_file.f90). I am not sure there is a diurnal variation at significant fire burning. Still, the authors should explain the diurnal variation used in this study, and I think the diurnal variation should be validated by observed diurnal cycles of AOD (or AOT of HIMAWARI-8).
**Response: The version of FINN used for this study is 1.5. As for the plume rise calculation, it is identical to HD18, and we will mention this in the revised manuscript.**

**The figure below shows the comparison of AOD diurnal cycles observed by AERONET (during the day, lime & green) and simulated (FIRE, red). Clearly, it is**

challenging to observe any diurnal cycle from the AERONET data as it uses a sun-photometer that can provide data only during the day. In addition, the AERONET AODs are clearly impacted by the sun angle at some stations (e.g., Fig d, f, g, h). The nighttime data unavailability is also the case for satellite AODs.

[Figure]

**Figure: Monthly mean AOD diurnal cycle observed by AERONET (green, averaged half-hourly) and simulated in the nearby grid box in the FIRE simulation (red, hourly). The shading ranges between the minimum and maximum AODs. Time (hours) is in UTC.**

3) Section 3.1, lines 129-131: I cannot understand why the authors insist "AOD measurement data from the nearby AERONET stations is unavailable." It seems several AERONET stations had AOD data in September 2015 (https://aeronet.gsfc.nasa.gov, I have accessed on 19 October 2021). For the conversion of 500nm AERONET AOD with 550nm AOD, equation 1 of Jiang et al. (2019; https://doi.org/10.3390/rs11091011) could be useful.

**Response: The authors have downloaded and checked the data before the initial submission, particularly the data of AOD at 532 nm and 551 nm for a fair comparison with the WRF-CHEM output (550 nm). These datasets are indeed missing at all the stations except for the one at GOT_Seaprism that we showed in the original manuscript. However, since we found that AOD data at 500 nm and 675 nm are available at all the stations except for GOT_Seaprism, we decided to additionally show them for all the other stations, in order to show the range of AOD values that is expected for 550 nm from the AERONET observations. Please see the new figure below. Thank you for your suggestion.**

[Figure]

**Figure:** (a) Locations of 9 AERONET stations whose AOD data for the month of September 2015 are shown in (b-j) in green (675 nm) and lime (500 nm). For (c), they are AODs at 551 nm (green) and 532 nm (lime) instead. Red and blue lines are estimated AODs at 550 nm from the FIRE and NOFIRE simulations, respectively.

Technical corrections:

1) Page 8, Figure 6: It seems the text "AOD from MODIS Aqua (MYD08_D3) shows a similar result" is not suitable for the figure caption. If there is no significant difference between Aqua and Terra, the authors should mention it in the main text.

**Response: We have modified the main text to "…observed by MODIS Terra (Figure 6a; Aqua shows a similar result)" so that this information is included in the main text.**

*Reference:*

**Lee et al. (2017): Biomass burning aerosols and the low-visibility events in Southeast Asia, Atmos. Chem. Phys., 17, 965–980, https://doi.org/10.5194/acp-17-965-2017**

---

## Author Response (AR1)

Authors' Response Pages 1-3: Response to Anonymous Referee #1 (RC3) Pages 4-9: Response to Anonymous Referee #2 (RC2)

Response to Anonymous Referee #1 (RC3)

**Thank you for your additional comments to our previous response (in italic font). Our responses to your comments are below (in bold blue).**

1. It would be helpful if the reviewer could provide more details on the data sources and their availability, so that we could further examine the quantitative differences between FINN v1.5 and GFED.

My previous figure is based on the output of FIRECAM (https://globalfires.earthengine.app/view/firecam) except for FINN2.4. I also made similar figures by myself, and I can not find a problem in my procedure. Please check the attached file (info\_for\_figure.zip). I am afraid that the authors have forgotten to divide by 30 (days) for the figure of FINN1.5.

Response: Thank you for sharing the files. We have carefully looked into the codes and realized that our previous figure for FINN was showing the mean PM2.5 emission rate averaged among fire cells (1 km2 per cell) within each  $0.1^{\circ} \times 0.1^{\circ}$  grid. However, there are a lot of cells that do not have any fires (i.e., simply no FINN data as there is no fire), which should also be included when calculating the  $0.1^{\circ} \times 0.1^{\circ}$  grid average. GFED provides this grid-mean data, so in order to have a fair comparison, we have re-calculated the PM2.5 emission rates in [ton/month/each  $0.25^{\circ} \times 0.25^{\circ}$  grid] as shown below; both of the plots match what was provided by the reviewer. Although this figure is not included in the paper, we appreciate the reviewer's comment, which has led to the clarification of the quantitative difference between FINN and GFED for the month of September in 2015.

**Figure:** Estimated PM2.5 emission rate [ton/month] in each 0.25° × 0.25° grid for the month of September in 2015, according to (left) FINN v1.5 and (right) GFED.

2. We deem that the ratio of 5:1 is not an uncommon parent-to-nested ratio and has been used in similar studies. It is not entirely clear to us what/which incorrect feedback (e.g., relevant to aerosol-cloud interaction?) was being referred to, and indeed, we would greatly appreciate it if the reviewer could provide some relevant publication on this matter. Even if the indexing was slightly off, we do not believe that it has an impact significant enough to invalidate our scientific findings from our simulations.

I am afraid that the authors have not checked the release note of WRF version 4.2 which I have shown, but detailed information related to this problem can be found at: https://github.com/wrf-model/WRF/pull/1100 I recommend the authors to conduct a sensitivity test by yourself whether this problem might affect your result or not, by using corrected source code (share/interp\_fcn.F).

Response: We have run a NOFIRE simulation with changes in the file (share/interp\_fcn.F) incorporated for the first 3 days of September 2015. Differences (NOFIRE - NOFIREnew) in temperature at the lowest model level is shown below; as is clear, the difference is very small, which indicates the negligible impact of the code change to our simulations results and therefore our findings.

---

## Author Response (AR2)

Response to Reviewer

In general, the authors addressed my comments well and I see significant improvements in the manuscripts. I noticed some corrections and clarifications are needed before the manuscript can be accepted.

**Thank you for your suggestions. We have modified the manuscript accordingly. Our responses to your comments are below. Additionally, we have made very minor modifications to the axis labels or captions of Figures 11, and A1-A3 for further clarification. We have added the following sentence to *Acknowledgements* as well.**

**Page 20 lines 309-310:**
*"… We also thank the two anonymous reviewers for their insightful comments and suggestions that allowed us to significantly improve the paper."*

**Moreover, we have added the following sentence in *Code and data availability* for further clarification;**

**Page 20 lines 299-300:**
*"Readers interested in the specific modifications made to the WRF-CHEM source code for this study can contact the corresponding author."*

1. WRF model was indeed coupled with ocean models in some studies (such as COAWST: https://www.sciencedirect.com/science/article/pii/S1463500310001113?via%3Dihub. COAWST code is publicly accessible). So it is not accurate to say "WRF/WRF-CHEM does not have the capability of coupling with an ocean model".
**Response: Thank you for pointing this out. We were unaware of the development. The text has been corrected as follows;**

**Page 7 lines 141-144:**
*" Some efforts have been already made to couple WRF/WRF-CHEM with an ocean model as in Warner et al. (2010) and Zhang et al. (2019). Over MC where the varying sea conditions can strongly influence convective activities, the use of such comprehensive models is more desirable and may lead to more realistic simulations."*

**Citation:**
1. **Zhang, Y., Wang, K., Jena, C., Paton-Walsh, C., Guérette, É.-A., Utembe, S., Silver, J. D., and Keywood, M.: Multiscale Applications of Two Online-Coupled Meteorology-Chemistry Models during Recent Field Campaigns in Australia, Part II: Comparison of WRF/Chem and WRF/Chem-ROMS and Impacts of Air-Sea**

Interactions and Boundary Conditions, Atmosphere, 10, https://doi.org/10.3390/atmos10040210, 2019b.

2. Warner, J. C., Armstrong, B., He, R., and Zambon, J. B.: Development of a Coupled Ocean–Atmosphere–Wave–Sediment Transport (COAWST) Modeling System, Ocean Modelling, 35, 230–244, https://doi.org/10.1016/j.ocemod.2010.07.010, 2010.

2. Saying "latent heat release from droplet activation" is not correct. Droplet activation is an instant process and does not involve much water phase condensation. I think you meant the droplet condensational growth after the activation. Stronger condensation in the polluted case brings down the supersaturation in updrafts, leading to lower supersaturation shown in the figure in the response letter, which would eb a nice figure to be included in the paper (can be a supplementary figure).

**Response: We used the expression of "latent heat release from droplet activation" in our previous response to the reviewer. We agree that this expression is inaccurate. In fact, what we really meant was "latent heat release following droplet activation". We have made sure that the latter expression is used in the manuscript. In addition, we have added the following figure of reduced supersaturation, which we have shown in our previous response to the reviewer, as Figure A4 in the revised manuscript along with its explanation in the main text.**

**Page 13 lines 213-214: "*Figure A3 shows the estimated amounts of maximum latent heat released  following droplet activation and freezing.*"**

[Figure]

**Figure A4: Differences (FIRE-NOFIRE) in maximum supersaturation S$_{max}$ averaged within each region, only sampled where updraft $\geq 5$ ms$^{-1}$ and S$_{max}$ > 0.**

**Page 13 lines 216-217: "*As a result of the enhanced condensation, the maximum supersaturation is lowered inside convection in FIRE (Figure A4).*"**

3. The autoconversion rate in the microphysics scheme you used is parameterized as decreasing with increased droplet number. So looking at the autoconversion rate alone is not adequately to say something about warm rain in your case. The best way is to track warm and melted rain in the code. If you do not want to address this since it involves in rerunning model simulations, that is fine since it is not a major point of the paper.

**Response: We agree with the reviewer that the autoconverstion rate alone is not sufficient as the representation of the entire warm rain processes. The rigorous separation of warm and cold rain indeed requires the modification of the source code and re-runs of the simulations, and therefore will remain as a future work.**

4. The reference is wrong in this sentence "Such aerosol-induced changes in stratiform anvil clouds, namely their extended lifetime and heightened cloud top, have been reported in previous studies (e.g., Fan et al., 2018)". It should be Fan et al., 2013.
**Response: The correction has been made in line 227 on Page 14.**